# Preclinical Pharmacokinetics and Bioavailability of Oxypeucedanin in Rats after Single Intravenous and Oral Administration

**DOI:** 10.3390/molecules27113570

**Published:** 2022-06-02

**Authors:** Ming-Cong Zheng, Wen-Ting Tang, Lu-Lu Yu, Xun-Jia Qian, Jie Ren, Jie-Jia Li, Wei-Wei Rong, Jun-Xu Li, Qing Zhu

**Affiliations:** 1School of Pharmacy, Nantong University, Nantong 226001, China; 1923310030@stmail.ntu.edu.cn (M.-C.Z.); 1923310010@stmail.ntu.edu.cn (W.-T.T.); 2019310026@stmail.ntu.edu.cn (L.-L.Y.); 2019310024@stmail.ntu.edu.cn (X.-J.Q.); 1827021016@stmail.ntu.edu.cn (J.R.); rongweiwei@ntu.edu.cn (W.-W.R.); junxuli@gmail.com (J.-X.L.); 2Provincial Key Laboratory of Inflammation and Molecular Drug Target, Nantong 226001, China; 3State Key Laboratory for the Quality Research of Chinese Medicine, School of Pharmacy, Macau University of Science and Technology, Macau 999078, China; lijiejia@ntu.edu.cn

**Keywords:** oxypeucedanin, pharmacokinetics, bioavailability, UPLC/MS/MS, rats

## Abstract

Oxypeucedanin, a furanocoumarin extracted from many traditional Chinese herbal medicines, has a variety of pharmacological effects. However, the independent pharmacokinetic characteristics and bioavailability of this compound remains elusive. In this study, a rapid, sensitive, and selective method using ultra-high performance liquid chromatography–tandem mass spectrometry (UPLC/MS/MS) was developed for evaluating the intravenous and oral pharmacokinetics of oxypeucedanin. After intravenous administration of oxypeucedanin (2.5, 5, and 10 mg/kg), and intragastric administration of oxypeucedanin (20 mg/kg), blood samples were collected periodically from the tail vein. The plasma concentration-time curves were plotted, and the pharmacokinetic parameters were calculated using a non-compartmental model analysis. After intravenous administration of oxypeucedanin (single dosing at 2.5, 5, and 10 mg/kg) to rats, the pharmacokinetics fit the linear kinetics characteristics, which showed that some parameters including average elimination half-life (T_1/2Z_ of 0.61~0.66 h), mean residence time (MRT of 0.62~0.80 h), apparent volume of distribution (V_Z_ of 4.98~7.50 L/kg), and systemic clearance (CL_Z_ of 5.64~8.55 L/kg/h) are dose-independent and the area under concentration-time curve (AUC) increased in a dose-proportional manner. Single oral administration of oxypeucedanin (20 mg/kg) showed poor and slow absorption with the mean time to reach the peak concentration (T_max_) of 3.38 h, MRT of 5.86 h, T_1/2Z_ of 2.94 h, and a mean absolute bioavailability of 10.26% in rats. These results provide critical information for a better understanding of the pharmacological effect of oxypeucedanin, which will facilitate its research and development.

## 1. Introduction

Oxypeucedanin (5-epoxy-isopentenyloxypsoralen, C_16_H_14_O_5_), a linear furanocoumarin, is the major component of some herbs (Umbelliferae family), such as *Angelica apaensis Shan et Yuan* and *Angelicae Dahurica* Radix (ADR) [1,2]. *Angelica apaensis* has been traditionally applied to the treatment of gastrointestinal pain, coughs, and headaches for several hundreds of years in Yunnan Province, China [3]. ADR (Bai-zhi in Chinese), an herbal medicine exhibiting antioxidative, anti-inflammatory, antiproliferative, anti-tumor, antimicrobial and anti-Alzheimer disease effects, has been widely used for the treatment of rheumatism and pain relief in China [4,5]. Notably, the root part of ADR has been characterized as the richest natural source of oxypeucedanin [6]. Previous studies have indicated that oxypeucedanin have various biological activities including antiarrhythmic activity, anti-oxidant, anti-inflammatory, analgesic, anti-cancer, and antibacterial effects [6,7,8,9]. Furthermore, oxypeucedanin was reported to be a potentially effective intervention for nephrotoxicity induced by sunitinib [10]. 

Pharmacokinetic study plays an important role in the early stage of the development of new drugs [11]. The failure of drug development is to a certain extent attributed to the poor pharmacokinetic properties, such as low bioavailability or low metabolic stability [12]. So far, some studies for pharmacokinetic profiles of several coumarins including oxypeucedanin after administration of the herb medicine ADR extract have been reported. One study reported that the elimination of oxypeucedanin was rapid, with elimination half-life (T_1/2Z_) of 1.6 h and mean residence time (MRT) of 2.3 h, after an intravenous dose of RAD extract in rats [13]. After oral administration of ADR extracts, two studies indicated that the absorption of oxypeucedanin was rapid, with the time to reach the peak concentration (T_max_) of 0.51 h [14] or 43.2–49.1 min [15]; however, another study obtained different results that oxypeucedanin displayed slow absorption with T_max_ of 12 h [13]; the tissue distribution test indicated that oxypeucedanin was rapidly and widely distributed into various tissues and lung is the most distributed tissue [16]; the excretion experiment showed that oxypeucedanin was cumulatively excreted in bile and urine at 0.17% and 0.082%, respectively [17]. After oral administration of Radix Glehniae (named Beishashen in China) extract, the cumulative percentages of oxypeucedanin excreted in the urine and bile over the dose administered were 0.034% and 0.039%, respectively [18]. In addition, oxypeucedanin was reported to improve the P-glycoprotein (P-gp) mediated drug transport and remarkably increase the absorption of the cancer medication docetaxel (the substrate of P-gp) [19].

However, the pharmacokinetic properties after administration of oxypeucedanin alone still have not been reported. Due to the complexity of ingredients in Chinese medicine and the presence of drug-drug interactions, the accurate pharmacokinetic studies of oxypeucedanin need to be performed. In the present study, pharmacokinetics of oxypeucedanin after single intravenous and oral administration and its absolute bioavailability in rats were investigated, which may shed light on the future development of oxypeucedanin as a drug candidate.

## 2. Materials and Methods

### 2.1. Chemicals and Reagents

Oxypeucedanin (purity > 99.0%) was purchased from Shanghai Ronghe Pharmaceutical Technology Development Co. Ltd. (Shanghai, China). The internal standard (IS), imperatorin (purity > 99.0%) was purchased from Aladdin (Shanghai, China). The chemical structures of oxypeucedanin and IS are shown in Figure 1. All other chemicals and reagents were HPLC-grade. Deionized water was prepared using a Milli-Q plus Ultra Pure water system (Millipore, Shanghai, China).

### 2.2. Animal Experiments

Adult male and female Sprague-Dawley rats weighing of 250–300 g (Laboratory Animal Center, Nantong University, Nantong, China) were housed in groups and habituated to the controlled laboratory environment (temperature of 22 ± 1 °C, relative humidity of 50–70%, and 12 h light/dark cycle, lights on at 7:00 a.m.) with free access to food and water for at least 7 days. The animal study was approved by the Institutional Animal Care and Use Committee, Nantong University. Animals were maintained in accordance with the Regulations of Experimental Animal Administration issued by the State Committee of Science and Technology of People’s Republic of China. At the beginning of each experiment, the rats were fasted for 12 h before dosing and 2 h afterwards, with free access to water. The dosing levels at 2.5, 5, and 10 mg/kg (with the volume of 5 mL/kg) were used for intravenous injection (*i.v.*) and 20 mg/kg (with the volume of 10 mL/kg) were used for intragastric administration (*i.g.*). The solution of oxypeucedanin for *i.v.* bolus was prepared as follows: firstly, a certain amount of oxypeucedanin was accurately weighed and dissolved into the mixture containing 100 μL ethanol and 100 μL PEG400 to obtain the solution 1; then 90 mg 2-hydroxypropyl-beta-cyclodextrin (HP-beta-CD) was dissolved in 800 μL water at 60 °C to obtain the solution 2; finally, the solution 2 was slowly dropped into the solution 1 and then was filtrated using 0.45 µm membrane to obtain the injection. The oxypeucedanin suspension prepared by 0.5% sodium carboxymethyl cellulose (CMC-Na) was used for oral administration. After administration, approximately 0.3 mL of blood sample was collected from each rat at time points of 0 (pre-dose), 0.033, 0.117, 0.25, 0.5, 1, 1.5, 2, 3, 4 h for the *i.v.* bolus group through the posterior orbital venous plexus and 0, 0.167, 0.5, 1, 1.5, 2, 3, 4, 6, 8, 10, 12 h for the *i.g.* group through the tail vein. According to the international guideline [20], the maximum volume of blood that can be collected may not exceed 1% of the body weight in any animal in a two-week period, which has been approved by the Institutional Animal Care and Use Committee of Nantong University. Thus, the normal saline 1 mL was injected intravenously to rats every three blood sampling points to maintain the internal blood volume. The blood sampling schedule was designed based on the results from our preliminary experiment. The plasma was separated by centrifuging at 4000× rpm at 4 °C for 10 min and then stored at −80 °C until analysis.

### 2.3. Analytical Method for Oxypeucedanin and Method Validation

The detection and quantification of analytes were performed on an ultra-high performance liquid chromatography–tandem mass spectrometry (UPLC/MS/MS) system consisting of an Agilent 1290 series UPLC system (Agilent Technologies, Palo Alto, CA, USA) and a SCIEX 5500 QTrap triple quadrupole/linear ion trap hybrid mass spectrometer equipped with an electrospray ionization (ESI) source (Applied Biosystems/MDS Sciex, Concord, ON, Canada). Data acquisition and processing were performed by Analyst 1.6.2 software (Applied Biosystems/MDS Sciex, Concord, ON, Canada).

An Eclipse Plus C18 (50 mm × 2.1 mm, 1.8 μm) maintained at 30 °C was used for chromatographic separation that was run by gradient elution with the mobile phase consisting of 0.1% formic acid in water (A) and acetonitrile (B) at the flow rate of 0.2 mL/min. A gradient elution were programmed as follows: 25–35% B, 0–1 min; 35–60% B, 1–4 min; 60–25% B, 4–6 min; and 25% B, 6–8 min. In the MS/MS analysis, positive ion mode and a multiple reactions monitoring (MRM) were used. Ion source parameters was programmed as follows: spray voltage (4.0 kV), gas ion source 1 (50 psi), gas ion source 2 (50 psi), Curtain Gas (40 psi), and temperature (350 °C). Declustering potential (DP, 171 V), collision energy (CE, 23 V), entrance potential (EP, 10 V) and collision exit potential (CXP, 13 V) were optimized for oxypeucedanin and DP (126 V), CE (17 V), EP (10 V), and CXP (10 V) were set for IS. Quantification was performed with multiple reactions monitoring (MRM) of precursor–product ion transitions of *m*/*z* 287→203 for oxypeucedanin and *m*/*z* 271→203 for IS. Rat plasma samples were thawed at room temperature before use. To a 50 μL aliquot of plasma standard or sample, 5 μL IS (1 μg/mL) and 150 μL methanol were added. After vortexing for 2 min and centrifugation (12,000 rpm) at 4 °C for 10 min, 5 μL supernatant were injected into the UPLC/MS /MS system.

The method validation was conducted for specificity, linearity, precision, accuracy, recovery, matrix effect, and stability according to the guidelines for Bioanalytical Method Validation of the Food and Drug Administration (US FDA, 2001). Quality control (QC) samples (*n* = 6) at three concentrations (6, 60, and 900 ng/mL) were used for method validation. The matrix effects were evaluated using the post-extraction spike method. The matrix effect (%) was expressed as the ratio of the mean peak area of analytes spiked in the blank plasma samples after protein precipitation to that of the neat standard analyte spiked in the solution mobile phase at equivalent concentrations.

### 2.4. Pharmacokinetic Calculations

The mean plasma concentration-time curve was plotted. The pharmacokinetic parameters including maximum plasma concentration (C_max_), the time to reach the peak concentration (T_max_), the area under the curve from time zero to the last sampling point (AUC_0–t_) and up to infinity (AUC_0–∞_), mean residence time (MRT), half-life (T_1/2Z_), systemic clearance rate (CL_Z_, L/kg/h) and apparent volume of distribution (V_Z_, L/kg) were calculated by a non-compartmental model by the pharmacokinetic software, DAS 2.0 (issued by the State Food and Drug Administration of China). Bioavailability (F) after oral administration was calculated by the following formula:$$F=\frac{{\mathrm{Dose}}_{i.v.}\times {\;\mathrm{AUC}}_{i.g.}}{{\mathrm{Dose}}_{i.g.}\times {\;\mathrm{AUC}}_{i.v.}}\times 100\%$$
where Dose*_i.g._*: the oral administration dose; Dose*_i.v._*: the intravenous dose; AUC*_i.g._*: the AUC_0–∞_ after oral administration; AUC*_i.v._*: the AUC_0–∞_ after intravenous injection.

### 2.5. Statistical Analysis

All values are presented as means ± standard deviation. The estimated pharmacokinetic parameters from each treatment were statistically compared using one way ANOVA followed by Bonferroni’ *post hoc* analysis and the comparisons between male and female rats were performed using Student’s *t*-test by GraphPad Prism 8.0 (GraphPad Software, San Diego, CA, USA). The level of *p* < 0.05 was considered as statistical significance.

## 3. Results and Discussion

### 3.1. Method Development

In our work, imperatorin was used as IS because it has similar chemical structure to oxypeucedanin, which have both furan ring and α-benzopyrone [13,15]. Based on mass spectrometry full scan (Q1) of oxypeucedanin and IS and the product ion fragments in product ion scan (MS2) mode, we selected multiple-reaction monitoring (MRM) of precursor–product ion transitions with *m*/*z* 287→203 for oxypeucedanin and *m*/*z* 271→203 for IS (Figure 1).

The Eclipse Plus C18 column (50 mm × 2.1 mm, 1.8 μm) was selected as the stationary phase to provide better sensitivity and peak shape. In the mobile phase optimization, acetonitrile was selected as the organic phase, because it showed better elution performance and less noise and interference than methanol. Using 0.1% formic acid water as water phase can reduce tailing. Under the gradient elution of 0.1% formic acid water and acetonitrile, desired separation of oxypeucedanin and imperatorin (IS) was achieved with the retention time 4.4 ± 0.1 min for oxypeucedanin and 5.5 ± 0.1 min for IS, as shown in Figure 2. The total running time was 8 min, suggesting that this analytical method was fast and could be used for the determination of analytes in large quantities.

### 3.2. Method Validation

The representative chromatograms of blank plasma sample, plasma sample spiked with oxypeucedanin at the lower limits of quantification (LLOQ) level and IS, and the in vivo plasma sample collected from rats after the oral administration of oxypeucedanin were shown in Figure 2. Under analytical conditions, a tiny endogenous interference peak was found in blank plasma at each retention time of oxypeucedanin and IS, but its area was less than 10% of LLOQ and 5% of IS respectively. This showed that the analytical method had good specificity and selectivity. The linear range of the calibration curves was 2–1000 ng/mL for oxypeucedanin. A typical regression equation was Y = 0.00858X + 0.00985 with a correlation coefficient (r) of 0.999 using weighting factor (1/x^2^). The linear relationship between concentration and response was well linear with a correlation coefficient (r) ≥ 0.995 in all batches analyzed (*n* = 6). The LLOQ for this assay was validated at 2 ng/mL with signal-to-noise ratios (S/N) ≥ 10 in plasma, which was sufficiently sensitive to measure the concentration of the last sampling point.

Table 1 showed that the accuracy of all calibrator standard plasma samples ranged from 87.05% to 106.55%, and the precision (expressed as percent form of relative standard deviation (%RSD)) was lower than 15% except for LLOQ (2 ng/mL) that was less than 20%. The precision and accuracy of the calibration curves are acceptable.

The results of the assay precision and accuracy evaluated at three QC levels were summarized in Table 2. The intra- and inter-day assay accuracy of all analytes ranged from 92.51% to 106.98%, and the intra- and inter-day assay precision was 4.77–8.22%. They all met the acceptable criteria.

The results of autosampler, short-term, long-term, and freeze–thaw stability were shown in Table 3. The accuracy of QC samples for the stability experiment was in the range of 85–115% with the precision below 15%. It indicated that oxypeucedanin in rat plasma was stable under our experimental conditions.

Table 4 showed that the mean extraction recoveries for all QC samples were 92.52–103.8% with precision 3.13–6.02%, and their matrix effects were 98.02–109.03% with precision 3.92–11.52%. The results showed that the extraction efficiency was satisfactory, and the recovery was stable and reproducible at various concentrations. There was no significant matrix effect. 

When the in vivo plasma concentration exceeded the upper limit quantification (1000 ng/mL) in this analytical method, the appropriate amount of blank rat plasma was required to dilute this sample and the reliability of dilution effect was investigated. The accuracy and precision of the dilution effect were calculated as 98.64% and 4.94%, respectively, which met the requirements of determination. In addition, no chromatographic carry-over effect was observed injecting a blank sample after the highest standard calibration point. 

All these results complied with the FDA guidance of bioanalytical method validation. In summary, we developed a sensitive, rapid, and specific UPLC-MS/MS method for the determination of oxypeucedanin in pharmacokinetic studies in rats.

### 3.3. Pharmacokinetics and Bioavailability of Oxypeucedanin

This UPLC-MS/MS method was applied successfully to the pharmacokinetic study of oxypeucedanin in healthy adult rats after intravenous and oral administration. Our research group found that oxypeucedanin shows analgesic effects for chronic pain conditions. We found that single intraperitoneal (i.p.) injection of oxypeucedanin 5–20 mg/kg significantly attenuated the mechanical allodynia in a dose-dependent manner in chronic constricted injury (CCI) induced neuropathic pain model rats and repeated administration can maintain this effect without the trend of analgesic tolerance. Oxypeucedanin 10 mg/kg (i.p.) has already exhibited the maximal effect in this condition (see Appendix A). Thus, we selected the three dose levels at 2.5, 5, and 10 mg/kg for the intravenous pharmacokinetic experiments. Due to the previous report that the bioavailability of oxypeucedanin was about 10% after oral administration of ADR extract [13], we chose the dose 20 mg/kg for the intragastric administration. The mean plasma concentration-time curves of oxypeucedanin after intravenous and oral administration were shown in Figure 3, and the mean pharmacokinetic parameters obtained from the plasma concentration-time data were shown in Table 5. For all the concentration-time profiles in the study, the ratio of AUC_0–t_ to AUC_0–∞_ was more than 90% and the concentration of the last sampling point was less than 10% of the C_max_, indicating that AUC_0–∞_ was reliable and the sampling schedule was appropriate.

Following *i.v.* administration, the concentration of oxypeucedanin declined quickly and the elimination half-lives of oxypeucedanin ranged from 0.25 to 0.88 h, indicating that oxypeucedanin underwent rapid distribution in the whole body and then was quickly eliminated. The pharmacokinetic profiles after *i.**v.* administration at dose of 2.5, 5, and 10 mg/kg showed similar average elimination half-life (T_1/2Z_ of 0.61~0.66 h), mean residence time (MRT of 0.62~0.80 h), volume of distribution (V_Z_ of 4.98~7.50 L/kg), and systemic clearance (CL_Z_ of 5.64~8.55 L/kg/h), which have no differences between different dose levels (*p* > 0.05). The AUC_0–t_ increased in a dose-proportional manner (see Appendix A). These results indicated that the elimination of oxypeucedanin fits the linear kinetics characteristics. Furthermore, the volume of distribution (V_Z_) was much higher than the total body water volume (0.6 L/kg) in the rats [13], indicating an extensive extravascular uptake of oxypeucedanin. It may widely distribute in blood, as well as in many other tissues, which might contribute to its broad biological activities in various tissues. The clearance (CL_Z_) of oxypeucedanin was about two-fold that of the hepatic flow velocity of the rat liver (3.3 L/kg/h) [13], which suggested that oxypeucedanin was quickly cleared in rats, and its extrahepatic clearance pathways including renal clearance may also importantly contribute to the clearance of this compound, except for the hepatic metabolism as the main clearance pathway. Previous study had reported the pharmacokinetic parameters of oxypeucedanin after *i.v.* administration of ADR extract, which were different from the results in the present study [13]. These discrepancies may be attributed to the fact that the elimination of oxypeucedanin was inhibited or reduced by some coumarin components in ADR extract, which resulted in the prolongation of T_1/2Z_ and MRT and the decrease of CL_Z_. For example, the two furanocoumarins in ADR extract, psoralen [21] and xanthotoxol [22] were reported to concentration-dependently inhibit the activity of CYP3A4 that may be responsible for the metabolism of oxypeucedanin [23].

Following *i.g.* administration, the absorption of oxypeucedanin appears to be slow due to the longish T_max_ (3.38 ± 0.74 h) and delayed MRT (5.86 ± 2.24 h) compared to that after *i.v.* administration. Previous reports showed conflicting results that from two studies the absorption of oxypeucedanin was rapid (T_max_ of 0.51 h or 43.2–49.1 min) [14,15] but from another study oxypeucedanin displayed slow absorption with T_max_ of 12 h [13] after oral administration of the ADR extract. These inconsistent findings may be attributed to the different dose levels and the drug-drug interactions among the several furanocoumarins [16]. The absolute bioavailability of oxypeucedanin after single dose (20 mg/kg) was about 10.26 ± 2.02%, indicating the poor absorption of oxypeucedanin in rats, which is consistent with the previous report [13]. In addition, no gender difference was found in the pharmacokinetic profiles after intravenous or oral administration of oxypeucedanin within the dose levels used in the present study (see the Appendix A).

## 4. Conclusions

For the first time, we have reported the accurate pharmacokinetic profile of oxypeucedanin and its absolute bioavailability in rats after single intravenous and oral administration of this compound alone, on the basis of development of a sensitive, accurate, and precise UPLC/MS/MS method for the determination of oxypeucedanin in plasma. These results provide essential information for a better understanding of the pharmacological effect of oxypeucedanin, which will facilitate the research and development of oxypeucedanin as a new drug candidate.

## Figures and Tables

**Figure 1 molecules-27-03570-f001:**
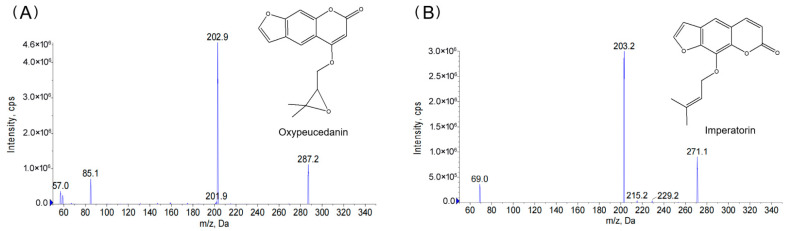
The chemical structure and the product ion spectra of oxypeucedanin (**A**) and the internal standard imperatorin (**B**) with monitoring at *m*/*z* 287→203 for oxypeucedanin and 271→203 for imperatorin (IS).

**Figure 2 molecules-27-03570-f002:**
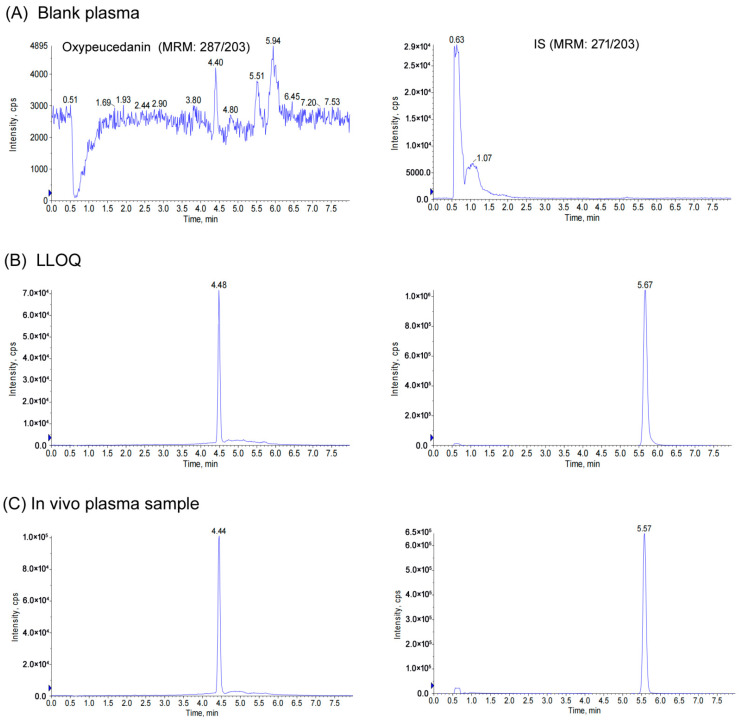
Representative MRM chromatograms of oxypeucedanin and IS in rat plasma: (**A**) blank plasma; (**B**) blank plasma spiked with oxypeucedanin at LLOQ level and IS; (**C**) an in vivo plasma sample collected at 12 h after oral administration of oxypeucedanin in rat at a dose of 20 mg/kg.

**Figure 3 molecules-27-03570-f003:**
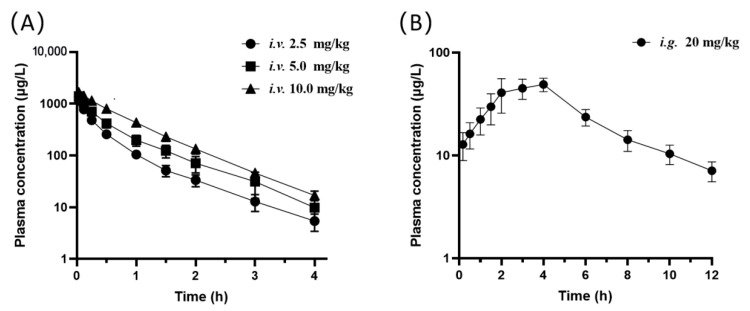
The mean plasma concentration-time curves of oxypeucedanin in log scale of ordinate (*n* = 8): (**A**) *i.v.* administration of oxypeucedanin at 2.5, 5, and 10 mg/kg; and (**B**) *i.g.* administration of oxypeucedanin at 20 mg/kg.

**Table 1 molecules-27-03570-t001:** Accuracy and precision in the calibration curve of oxypeucedanin.

Nominal Concentration (ng/mL)	Determined Concentration (ng/mL)	Accuracy (%)	Precision (%)
2(LLOQ)	1.84 ± 0.29	87.05	16.19
5	5.4092 ± 0.59	106.55	10.98
10	9.7690 ± 0.86	94.91	8.77
50	53.5744 ± 5.88	95.71	10.97
100	114.2086 ± 10.72	104.98	9.39
500	513.9346 ± 17.32	101.61	3.37
1000	968.5228 ± 31.61	98.98	3.26

**Table 2 molecules-27-03570-t002:** Precision and accuracy of oxypeucedanin in rat plasma (intra-day, *n* = 5; inter-day, *n* = 15, 3 days).

Nominal Concentration (ng/mL)	Intra-Day (*n* = 5)			Inter-Day (*n* = 15)		
	Determined Concentration (ng/mL)	Accuracy (%)	Precision (%)	Determined Concentration (ng/mL)	Accuracy (%)	Precision (%)
2(LLOQ)	2.00 ± 0.24	100.87	12.19	1.92 ± 0.27	96.25	13.86
6	6.02 ± 0.37	106.98	6.19	6.36 ± 0.49	106.48	7.71
60	57.05 ± 2.62	98.75	4.61	59.09 ± 4.86	98.49	8.22
900	861.82 ± 22.42	97.24	2.60	832.55 ± 39.68	92.51	4.77

**Table 3 molecules-27-03570-t003:** Stability data for oxypeucedanin in rat plasma under different conditions (*n* = 5).

Conditions	Determined Concentration (ng/mL)	Accuracy (%)	Precision (%)
6 ng/mL			
Autosampler (room temperature, 12 h)	5.72 ± 0.45	95.39	7.91
Bench-top (room temperature, 12 h)	5.98 ± 0.68	99.67	11.40
Freeze–thaw (three cycles)	6.14 ± 0.22	102.36	3.52
Long-term (1 month at −80 °C)	5.73 ± 0.32	95.50	5.65
60 ng/mL			
Autosampler (room temperature, 12 h)	66.76 ± 1.33	111.26	1.99
Bench-top (room temperature, 12 h)	62.56 ± 1.66	104.26	2.66
Freeze–thaw (three cycles)	59.87 ± 2.19	99.79	3.66
Long-term (1 month at −80 °C)	63.12 ± 2.33	105.20	3.69
900 ng/mL			
Autosampler (room temperature, 12 h)	923.85 ± 38.89	102.65	4.21
Bench-top (room temperature, 12 h)	943.02 ± 52.15	104.78	5.53
Freeze–thaw (three cycles)	873.81 ± 16.52	97.09	1.89
Long-term (1 month at −80 °C)	870.48 ± 68.68	96.72	7.89

**Table 4 molecules-27-03570-t004:** Recovery and matrix effect for oxypeucedanin in rat plasma (*n* = 3).

QC Samples	Recovery(%)	Precision for Recovery (RSD %)	Matrix Effect (%)	Precision for Matrix Effect (RSD %)
6 ng/mL	103.68	6.02	109.03	3.92
60 ng/mL	92.52	4.79	99.86	10.13
900 ng/mL	99.76	3.13	98.02	11.52

**Table 5 molecules-27-03570-t005:** The estimated mean pharmacokinetic parameters of oxypeucedanin after intravenous and oral administration (*n* = 8).

Parameters	*i.v.*			*i.g.*
	2.5 mg/kg	5 mg/kg	10 mg/kg	20 mg/kg
C_max_ (μg/L)	-	-	-	64.64 ± 34.79
C_2min_ (μg/L)	1140.35 ± 477.81	1393.22 ± 800.06	1662.94 ± 229.57	-
T_max_ (h)	-	-	-	3.38 ± 0.74
T_1/2Z_ (h)	0.61 ± 0.18	0.66 ± 0.13	0.61 ± 0.18	2.94 ± 1.59
AUC_(0–t)_ (μg/L∙h)	479.75 ± 142.84	760.67 ± 414.15	1282.04 ± 328.82	284.91 ± 87.29
AUC_(0–∞)_ (μg/L∙h)	484.47 ± 143.86	771.31 ± 425.43	1298.92 ± 332.04	318.25 ± 98.27
AUC_(0–t)_/AUC_(0–∞)_	0.99 ± 0.01	0.99 ± 0.01	0.99 ± 0.01	0.90 ± 0.09
MRT (h)	0.62 ± 0.27	0.72 ± 0.25	0.80 ± 0.19	5.86 ± 2.24
V_Z_ (L/kg)	4.98 ± 2.33	7.50 ± 3.32	7.04 ± 2.50	-
CL_Z_ (L/kg/h)	5.64 ± 1.90	8.55 ± 4.95	8.18 ± 2.22	-
F (%)	-	-	-	10.26 ± 2.02

C_max_, peak plasma concentration; C_2 min_, the plasma concentration at 2 min after *i.v.* injection; T_max_, the time to reach the peak concentration; AUC_0–t_, the area under the curve from time zero to the last sampling point; AUC_0–∞_, the area under the curve from time zero to infinity; MRT, mean residence time; T_1/2Z_, elimination half-life; CL_Z_, clearance; V_Z_, apparent volume of distribution; F, bioavailability.

## Data Availability

Not applicable.

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
