# Peer review of "Preclinical Pharmacokinetics and Bioavailability of Oxypeucedanin in Rats after Single Intravenous and Oral Administration"

_molecules, 2022, doi:10.3390/molecules27113570_

Round 1

Reviewer 1 Report

The paragraph "Recently our research group also found that oxypeucedanin shows analgesic effects for chronic pain conditions (data not shown), suggesting that oxypeucedanin may be further developed as a novel agent for the treatment of chronic pain." does not report any novel information. As authors describe previouly, "Previous studies have indicated that oxypeu-cedanin have various biological activities including anti-oxidant, anti-inflammatory, analgesic, anti-cancer, and antibacterial effects. [6-9]."

The sentence "Pharmacokinetic study plays an important role in the early stage of development of new drug." needs references.

The sentence "The failure of drug development is to a certain extent attributed to the poor pharmacokinetic prop-erties, such as low bioavailability or low metabolic stability." needs references

Overall, the work needs more introduction and references. 

Authors describe "approximately 0.3 ml of blood sample was collected" at different time points. Did the authors injected saline/PBS in order to maintain the internal blood volume? If not, How did the authors calculate the rat blood volume?. Authors can be overestimating the concentration of drug in blood. Moreover, authors extracted 1.5 ml of blood in 0.5h, which is around 10% of the total blood volume and 3 ml (20%) at the end point of the experiment (4h). Are these values approved by the Institutional Animal Care and Use Committee, Nantong University? International Guideline recommends "The maximum volume of blood that can be collected may not exceed 1% of the body weight (or 10 ml/kg) in any
animal in a two‐week period." In addition, alternatives to the retro-orbital plexus should be used whenever possible. Please see this information:
https://animal.research.uiowa.edu/iacuc-guidelines-blood-collection

Reviewer 2 Report

In this manuscript, the authors provided pharmacokinetics and bioavailability of oxypeucedanin in rats after single intravenous and oral administration.  The data were scientifically sound, and should be useful information for the further utilization of oxypeucedanin. Some suggestion and comments were listed below for the consideration of revision.

  1. In the introduction, the authors mentioned the research group found that oxypeucedanin shows analgesic effects for chronic pain conditions. Authors are suggested to provide comparison of the effective dose for chronic pain conditions with that for this study. Please provide some descriptions or supplementary data to clarify this viewpoint.
  2. The authors provided different ways to estimate the pharmacokinetic characteristics and bioavailability of oxypeucedanin administration. Please add some descriptions on why the authors chose 20 mg/kg for intra-gastric administration? Did the authors estimate higher or lower doses in the study?
  3. To estimate the pharmacokinetic properties after administration of oxypeucedanin alone was the aim of the study. The time point of blood sample collection was different from the other papers (References 10-12). Please provide some description of why the authors chose these time points.
  4. Please add the equation of Matrix effect in Materials and methods.
